# Factors Associated with Intention to Receive Vaccination against COVID-19 in Puerto Rico: An Online Survey of Adults

**DOI:** 10.3390/ijerph18157743

**Published:** 2021-07-21

**Authors:** Kyle Melin, Cheyu Zhang, Juan P. Zapata, Yonaira M. Rivera, Katie Fernandez, Enbal Shacham, Souhail M. Malavé-Rivera, Carlos E. Rodriguez-Diaz

**Affiliations:** 1Department of Pharmacy Practice, School of Pharmacy, University of Puerto Rico Medical Sciences Campus, San Juan, PR 00931, USA; 2Department of Prevention and Community Health, Milken Institute School of Public Health, The George Washington University, Washington, DC 20052, USA; sarahzhang@gwmail.gwu.edu (C.Z.); fernandezk@gwu.edu (K.F.); carlosrd@gwu.edu (C.E.R.-D.); 3Department of Psychology, Marquette University, Milwaukee, WI 53233, USA; juan.zapata@marquette.edu; 4Department of Communication, School of Communication and Information, Rutgers University, New Brunswick, NJ 08901, USA; yonaira.rivera@rutgers.edu; 5College for Public Health and Social Justice, Saint Louis University, St. Louis, MO 63104, USA; enbal.shacham@slu.edu; 6Center for Evaluation and Sociomedical Research, Graduate School of Public Health, Medical Sciences Campus, San Juan, PR P.O. Box 35607, USA; souhail.malave@upr.edu

**Keywords:** COVID-19, SARS-CoV-2, vaccine, immunization

## Abstract

We conducted an online survey among adults in Puerto Rico to identify factors associated with the intention to receive vaccination against COVID-19. Sociodemographic variables were analyzed independently for association with intent to receive vaccination. Significant associations were included in the multivariate logistic regression analysis. A total of 1016 responses were available for analysis. In the bivariate analysis, younger age, higher education, pre-COVID-19 employment, male sex, gay/bisexual identity, and single marital status were associated with increased intent to receive the vaccination. In the multivariate logistic regression, younger, male respondents, and those with higher educational attainment reported higher intent to receive the vaccination. Lower-income and living outside the San Juan metro region were associated with lower intent to receive the vaccination. National and international health organizations were identified as the most reliable sources of information, followed by healthcare professionals. These findings highlight the importance of considering sociodemographic characteristics and using trusted sources of information when designing COVID-19 vaccination public messaging.

## 1. Introduction

As of writing this report, the United States Centers for Disease Control and Prevention (CDC) has recommended the use of three Coronavirus disease 2019 (COVID-19) vaccines, with more in development [1]. Vaccination campaigns for COVID-19 will be key in curbing this global pandemic, but must be purposeful to ensure they do not propagate existing disparities of COVID-19′s impact on communities of color. COVID-19 has been particularly devastating to Black and Latino/Hispanic (henceforth: Latinx) communities in the U.S. [2,3]. Long-standing social inequities in social determinants of health place communities of color at elevated risk of COVID-19 infection, severe illness, and death compared to non-Hispanic whites [4]. Therefore, it is imperative to document and strategize how COVID-19 vaccines can be successfully disseminated throughout these communities.

Previous research shows that Latinx adults are less likely than non-Hispanic whites to be vaccinated [5,6]. Low perceived risk of influenza and healthcare providers’ failure to recommend vaccination may contribute to lower vaccination rates [7]. Compared to non-Hispanic whites, U.S. Latinxs reported more vaccine-related side effects and were more likely to be misinformed about the availability of vaccines [8]. In addition to individual-level barriers, systemic barriers have been well documented. Latinxs in the U.S. are less likely to have access to health care and more likely to suffer the effects of discriminatory policies, higher healthcare costs, lack of insurance, and language barriers; all of these factors constitute significant challenges to equitable public health efforts [3].

Data on the acceptability of the COVID-19 vaccines among minority populations are limited. However, a cross-sectional survey of a representative sample of U.S. adults found that both Black and Latinx respondents reported lower intent to be vaccinated for COVID-19 compared to non-Hispanic whites [9]. Another nationally representative longitudinal study found that intent to receive vaccination declined between April and December of 2020, with the highest decline observed among U.S. Hispanics [10]. This is consistent with previous findings from the 2009 H1N1 pandemic, which observed that intent to receive vaccination may be highest early in a pandemic before a vaccine is available [11].

With respect to the impact of COVID-19 in Puerto Rico (PR), it is important to highlight the unique socio-political status of PR as a U.S. territory, the economic crisis in 2016, and the devastating force of Hurricanes Irma and Maria in 2017 [12]. Because of these variables, and their ongoing impact in PR, it has been difficult to measure the impact of COVID-19 in PR as well as the overall effectiveness of current vaccination efforts. Current analyses suggest that the number of COVID-19 associated deaths in PR during the pandemic is higher than expected, with the number of COVID-19-associated deaths rising to more than double the numbers originally reported [13]. With the introduction of the SARS-CoV vaccines, it is important to determine variables associated with intent to receive vaccination. Unfortunately, there are no studies to date that have documented intent to receive vaccination in PR against COVID-19. In this Short Communication, we report the factors associated with the intent to receive vaccination against COVID-19 among adults living in Puerto Rico.

## 2. Materials and Methods

We conducted an online survey among adults 21 years of age and older living in Puerto Rico to identify social and behavioral factors impacted by the COVID-19 pandemic (21 is the age of majority in Puerto Rico). The data collection instrument was available in Spanish (Puerto Rico’s official language) in a secured online platform. The data collection instrument included several measures to assess socio-demographics, respondents’ experiences with COVID-19 information sources, perceived impact of COVID-19, and perceived governmental response, and were culturally adapted following a process previously used by the research team [14,15] and integrated previously validated measures [16]. The final survey instrument contained 70 items in total, and skip patterns were included to reduce burden among participants. Intention to get vaccinated against COVID-19 was assessed asking, “If there existed an approved vaccine to prevent the coronavirus infection, how likely is it that you would get vaccinated?” followed by a 4-point scale (not at all likely to very likely). Respondents were recruited using social media platforms and with the support of several community-based organizations that sent blast emails and posted study information on their websites and social media from 7 July 2020 to 13 July 2020. After screening and providing consent to participate, a total of *n* = 1031 respondents completed the survey. All research procedures were approved by the Human Research Subjects Protection Office of HIDDEN FOR PEER REVIEW.

Of the 1031 participants who completed the survey, two participants were excluded for incomplete data, leaving 1029 survey responses in the dataset. A total of 1016 responses were available for analysis from respondents who reported their intention to get vaccinated for COVID-19. The intent to receive vaccination was then dichotomized to maximize the number of participants in each category. Risk factor count was calculated as the number of the following factors: diagnosed with heart disease, respiratory illness, cerebral illness, diabetes, kidney disease, cancer, or age ≥ 65 years old. Health regions were derived from self-reported postcodes. All measured variables were selected a priori to be analyzed independently for association with intent to receive vaccination using Chi-square tests for categorical variables and t-tests for continuous variables. Variables with significant association and variables selected a priori (perception of government response and overall impact of COVID-19) were included in the multivariate logistic regression analysis. Adjusted odds ratios and their *p*-values for the independent variables were reported. The significance level of all analyses was at α = 0.05. All analysis was conducted with SAS 9.4 software.

## 3. Results

Respondents had a broad distribution of demographic and socioeconomic characteristics (Table 1), with a mean age of 46 years old and 73% identifying as women. In the bivariate analysis, younger age, higher education, pre-COVID-19 employment, male sex, gay/bisexual identity, and single marital status were all associated with increased intention to be vaccinated. Increasing numbers of risk factors for poorer outcomes from COVID-19 (i.e., aged 65 and older, comorbidities) were not associated with vaccine intention. 

No significant relationship was observed between pre-COVID-19 employment, sexual orientation, and marital status and intent to receive vaccination after adjustment (Table 1). However, younger, male respondents, and those with higher educational attainment, reported higher intent to receive vaccination. Respondents who had a yearly income of $50,000 or more had significantly higher intent to receive vaccination (POR = 2.4, *p* = 0.0096) than those who earned less than $10,000. Respondents living outside of the San Juan metro region had significantly lower intent to receive vaccination when compared to respondents who lived in the metro region (POR = 0.647, *p* = 0.0082). Respondents who were less likely to get vaccinated expressed less worry about the impact of COVID-19 and perceived the Puerto Rican government’s response to be deficient.

Respondents who reported a high degree of confidence in one’s ability to evaluate the credibility of information about COVID-19 reported higher intent to receive vaccination (Table 2). National and international health organizations (World Health Organization, Centers for Disease Control and Prevention) were identified as the most reliable sources for COVID-19-related information for respondents with a high likelihood of getting vaccinated (75%), followed by healthcare professionals (70%) and news programs (51%). While these sources were also selected as the most reliable among respondents with a low likelihood of vaccination, healthcare professionals and national/international health organizations were trusted at significantly lower rates than those intending to vaccinate (59% and 56%, respectively).

## 4. Discussion

These results, which are part of a larger survey created to evaluate the impact of COVID-19 on people living in Puerto Rico, provide insight into vaccine acceptability and intent to receive vaccination amongst the respondents and may be useful to current vaccination efforts in Puerto Rico and the U.S. Although the total number of risk factors for poor COVID-19 outcomes was not associated with intent to receive vaccination, decreased intention observed among older respondents is troubling.

Several limitations should be considered when interpreting the findings of this study. As mentioned previously, vaccine acceptability tends to decrease over time following the onset of a pandemic, and responses from July of 2020 may overestimate intent to receive vaccination. Moreover, the item assessing intent to receive vaccination was posed in a hypothetical manner as no vaccine was yet available. As such, perceptions and intentions may have changed since that time, in particular with regards to media coverage of different vaccine manufacturers and vaccines currently approved or in development. Further, the study used a convenient online sample of adults. This may have led to self-selected participants who had access to the internet and were interested in the topic. Despite these limitations, this was the best approach to capture information during the stay-at-home orders that were in place at the time of data collection. Lastly, it is possible that confounding factors not collected by the survey instrument such as religion or health literacy may have additional effects on intent to receive vaccination.

## 5. Conclusions

These findings highlight the importance of considering sociodemographic characteristics identified with low intent to receive vaccination when designing public messaging related to increasing COVID-19 vaccinations. In particular, older individuals earning less than $50,000 a year and those living in rural areas may be at particular risk of not being vaccinated. Furthermore, in light of reported perceptions of the government’s response to the COVID-19 crisis, communication initiatives should consider efforts to promote vaccinations via trusted sources of information, particularly among those who are less likely to get vaccinated. Communication campaigns may benefit from using established guidance from the CDC and WHO, and by being delivered by local healthcare professionals and through news outlets to ensure success.

## Figures and Tables

**Table 1 ijerph-18-07743-t001:** Bivariate and Multivariate Analyses of Socio-Demographic Characteristics with Vaccine Acceptability.

Variable (N, %)		If There Existed an Approved Vaccine to Prevent the Coronavirus Infection, How Likely is It that You Would Get Vaccinated?			Multivariate Regression Analysis	
		High Likelihood	Low Likelihood	*p*-Value	AOR	*p*-Value
Total participants (N, %)		704 (68.4%)	312 (30.3%)			
Age	*n*	704	312			
	Mean (SD)	44.9 (15.8)	50.1 (13.2)	<0.0001	0.98	0.0141
Age group		(*n* = 1016)		<0.0001		
21–29		180 (84.9%)	32 (15.1%)			
30–39		93 (73.8%)	33 (26.2%)			
40–49		121 (64.7%)	66 (35.3%)			
50–59		168 (63.4%)	97 (36.6%)			
60+		142 (62.8%)	84 (37.2%)			
Education		(*n* = 1013)		0.0013		
High school or less		35 (49.3%)	36 (50.7%)		(ref)	(ref)
Some years of university		161 (68.5%)	74 (31.5%)		2.168	0.0121
Bachelor’s degree		265 (70.3%)	112 (29.7%)		1.839	0.044
Post-graduate		241 (73.0%)	89 (27.0%)		1.952	0.0349
Refuse to answer		2 (66.7%)	1 (33.3%)		3.12	0.3949
Employment before COVID-19		(*n* = 1003)		0.0003		
Full time job		298 (71.5%)	119 (28.5%)		(ref)	(ref)
Part time job		86 (72.9%)	32 (27.1%)		1.008	0.9791
Student		83 (85.6%)	14 (14.4%)		1.978	0.0748
Retired		111 (64.2%)	62 (35.8%)		1.498	0.1147
Disabled		30 (56.6%)	23 (43.4%)		0.784	0.488
Unemployed		90 (62.1%)	55 (37.9%)		0.97	0.9108
Refuse to answer		6 (46.2%)	7 (53.8%)		0.902	0.8776
Employment under COVID-19		(*n* = 527)		0.2094		
The same (it has not changed after COVID-19 emergency)		297 (72.4%)	113 (27.6%)			
I lost my job during the emergency		51 (75.0%)	17 (25.0%)			
I’m laid off until further notice		30 (61.2%)	19 (38.8%)			
N/A		320 (66.5%)	161 (33.5%)			
Refuse to answer		6 (75.0%)	2 (25.0%)			
Income		(*n* = 970)		0.0201		
Less than $10,000		264 (70.6%)	110 (29.4%)		(ref)	(ref)
$10,000-$19,999		126 (68.1%)	59 (31.9%)		1.16	0.5313
$20,000-$34,999		145 (66.5%)	73 (33.5%)		0.945	0.8263
$35,000-$49,999		50 (64.1%)	28 (35.9%)		1.048	0.8908
$50,000 or more		95 (82.6%)	20 (17.4%)		2.677	0.0055
Refuse to answer		24 (52.2%)	22 (47.8%)		0.721	0.3936
What is your sex assigned at birth?		(*n* = 1016)		<0.0001		
Woman		496 (64.7%)	271 (35.3%)		(ref)	(ref)
Men		208 (83.5%)	41 (16.5%)		2.583	0.0001
What is your sexual orientation?		(*n* = 995)		<0.0001		
Heterosexual		525 (66.4%)	266 (33.6%)		(ref)	(ref)
Homosexual		112 (81.8%)	25 (18.2%)		0.895	0.7228
Bisexual		48 (85.7%)	8 (14.3%)		2.113	0.0747
Other		10 (90.9%)	1 (9.1%)		3.441	0.247
Refuse to answer		9 (42.9%)	12 (57.1%)		0.439	0.1165
What is your current marital status?		(*n* = 999)		<0.0001		
Single, never married		281 (79.2%)	74 (20.8%)		(ref)	(ref)
Currently married, living with partner		295 (66.6%)	148 (33.4%)		0.751	0.1863
Separated/divorced		99 (61.5%)	62 (38.5%)		0.708	0.1953
Widowed		20 (50.0%)	20 (50.0%)		0.458	0.0637
Refuse to answer		9 (52.9%)	8 (47.1%)		0.685	0.5258
Health region		(*n* = 937)		0.0002		
San Juan metro area		459 (73.8%)	163 (26.2%)		(ref)	(ref)
Other		195 (61.9%)	120 (38.1%)		0.647	0.0084
[Missing]		50 (63.3%)	29 (36.7%)		0.731	0.2849
Number of people living together	*n*	683	295			
	Mean (SD)	2.1 (1.5)	2.1 (1.5)	0.2164		
Number of children living together	*n*	656	278			
	Mean (SD)	0.5 (0.8)	0.6 (1.5)	0.0822		
Risk Factor Count		(*n* = 1011)		0.3365		
0		330 (71.0%)	135 (29.0%)			
1		200 (65.8%)	104 (34.2%)			
2		111 (72.1%)	43 (27.9%)			
3–6		58 (65.9%)	30 (34.1%)			
Refuse to answer		5 (100.0%)	0 (0.0%)			
Number of Risk Factors Reported	N	699	312			
	Mean (SD)	0.9 (1.1)	0.9 (1.1)	0.7107		
How do you consider the government of Puerto Rico’s response to the coronavirus pandemic?		(*n* = 1014)		0.0612		
Excellent/Good		147 (74.2%)	51 (25.8%)		(ref)	(ref)
Regular		274 (71.0%)	112 (29.0%)		0.636	0.0379
Bad/Very bad		282 (65.6%)	148 (34.4%)		0.372	<0.0001
Refuse to answer		1 (50.0%)	1 (50.0%)		0.633	0.757
How worried are you about the impact that COVID-19 can have on you?		(*n* = 1015)		<0.0001		
Extremely worried/Very worried		547 (72.1%)	212 (27.9%)		(ref)	(ref)
Somewhat worried		132 (67.3%)	64 (32.7%)		0.656	0.027
Just a little bit worried/I’m not worried		25 (41.7%)	35 (58.3%)		0.196	<0.0001
Refuse to answer		0 (0.0%)	1 (100.0%)		-	0.9791

Legend: AOR adjusted odds ratio; SD standard deviation.

**Table 2 ijerph-18-07743-t002:** COVID-19 Knowledge Acquisition and Perceptions of Vaccine Acceptability.

Variable (N, %)	If There Existed an Approved Vaccine to Prevent the Coronavirus Infection, How Likely is It that You Would Get Vaccinated?		
	High Likelihood	Low Likelihood	*p*-Value
How confident are you in your ability to determine if the information you are receiving about COVID-19 is correct?	(*n* = 1007)		0.0011
Super confident	288 (76.2%)	90 (23.8%)	
Somewhat confident	366 (65.5%)	193 (34.5%)	
Not confident at all	44 (62.9%)	26 (37.1%)	
Refuse to answer	6 (66.7%)	3 (33.3%)	
What sources do you trust to obtain reliable information regarding COVID-19? * (select all, *n* = 1016)			
Social media	303 (43.0%)	121 (38.8%)	0.2043
TV commercials	155 (22.0%)	63 (20.2%)	0.5134
News programs	358 (50.9%)	156 (50.0%)	0.8021
Peer network	115 (16.3%)	34 (10.9%)	0.0238
Healthcare professionals	495 (70.3%)	186 (59.6%)	0.0008
National/international health organizations	525 (74.6%)	174 (55.8%)	<0.0001
Local health department	349 (49.6%)	137 (43.9%)	0.0955
Other	39 (5.5%)	12 (3.8%)	0.2541
I don’t believe any source of information about COVID 19	3 (0.4%)	11 (3.5%)	0.0001
Refuse to answer	5 (0.7%)	5 (1.6%)	-

* Note: Respondents were able to select multiple options. Percentages noted for individual information sources are calculated by dividing the number of respondents who selected each information source divided by the total number of participants in the column (either high likelihood vs. low likelihood).

## Data Availability

Data are available upon completion of a data sharing agreement with the principal investigator, Carlos E. Rodriguez-Diaz.

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
