# Peer review of "Factors Associated with Intention to Receive Vaccination against COVID-19 in Puerto Rico: An Online Survey of Adults"

_ijerph, 2021, doi:10.3390/ijerph18157743_

Round 1
Reviewer 1 Report
A very interesting and timely manuscript. The reviewer has provided comments and follow up questions as an attachment for the authors' considerations. The author does not need to review the revised manuscript.

Reviewer 2 Report
Dear Author,
I consider this manuscript interesting, since thought this survey you have gathered a very good information about factors associated with intention to vaccine. However, this work has limitations in the methodology of survey.
The internet access is an important factor required to complete this online survey. Do you really think that you can obtain the same conclusions if the survey would be performed using other methodology?
My question is because in developing countries, internet is not essential and many people has not access, so in this study you cannot reach a wide part of the population, and the conclusions can be substantial different. The access to internet is in itself an excluding factor on this survey.
It would be interesting include a different way of survey to involve people with really low or absent incomes, places where the Covid left many victims. Furthermore, graphical representation of statistic results would be necessary.
Kind regards.
Reviewer 3 Report
The article submitted for review is of interest because it highlights the vaccination intentions of individuals in a country. However, the study method should be further specified. The authors should comment at greater length on where and how the response form was placed. I understand that it may be a web site, but it is of great interest to know what type of public potentially visits the form since it may generate a bias in the final results.
The number of subjects studied is also small and we do not know if it is a representative sample of the general population under study. These two data, for me, are necessary to clarify the validity of the study and the results obtained.
For this reason I would not accept the work in its current version.
Round 2
Reviewer 2 Report
Thank you for your explanations, but following your statement:
"While our study was not designed to be a truly representative sample of the general population, it is worth noting that the critical characteristics of the study sample are very similar with the sociodemographic characteristics of the American Community Survey in Puerto Rico"
If the study is not representative, this is a strong reason for biased results. I insist that the survey carried out under other conditions could be quite different. Despite the limitations due to Covid and sociodemographic conditions, the study could have been carried out with greater rigor. I highlight that the sociodemographic features are not a justification to perform a study in these conditions, in fact, this is the challenge.
Reviewer 3 Report
The article sent for review has significantly improved its quality. It is true that it deals with a local issue, but it may be of interest to other societies that can evaluate vaccination behavior.
I would accept the paper in its current version